# Different Fertility Approaches in Organic Hemp (*Cannabis sativa* L.) Production Alter Floral Biomass Yield but Not CBD:THC Ratio

Dylan Bruce, Grace Connelly and Shelby Ellison *

Department of Horticulture, University of Wisconsin-Madison, 1575 Linden Drive, Madison, WI 53706, USA; dbruce3@wisc.edu (D.B.); gconnelly@wisc.edu (G.C.)
* Correspondence: slrepinski@wisc.edu

**Abstract:** Industrial hemp is once again legal in the United States but agronomic practices are not well characterized, especially for organic production systems. Many producers are concerned that while increased fertility may increase biomass yield it will also disproportionally increase that rate of total tetrahydrocannabinol (THC) accumulation, making their crop more likely to be noncompliant at harvest. We measured the effect of four organic fertility treatments: (1) no fertility (NA); (2) manure-based compost (CM); (3) an industry-standard organic fertilizer blend (ST); and (4) an intensive fertility treatment designed for cannabis production (BQ), on biomass yield and cannabinoid accumulation over multiple timepoints and two years. We found that higher fertility (ST and BQ) led to increased biomass but did not influence the ratio of cannabidiol (CBD) to THC over time. We also found the ST treatment yielded the most consistently across years, whereas CM and BQ were less predictable and more costly, respectively. Our results show that fertility recommendations can be made to increase biomass yield without increasing the chance of noncompliance. Additional research addressing specific fertility requirements in organic cropping systems for hemp will be needed to meet consumer demand while remaining profitable and sustainable for growers.

**Keywords:** organic agriculture; *Cannabis sativa*; hemp; THC; CBD; fertility; compliance; biomass

## 1. Introduction

Hemp (*Cannabis sativa* L.) has regained popularity among growers due to excitement around potential consumer demand. Growers are interested in the prospective economic and agronomic benefits of having another option to add to the crop rotation on their farms. By US legal definition, hemp is a non-psychoactive subset of *Cannabis sativa* with less than 0.3% total tetrahydrocannabinol (THC) [1]. Hemp has many uses ranging from cannabinoid essential oils for supplement or pharmaceutical markets, to clothing, fuel, building materials, and food [2]. Currently, cannabinoid production is one of the most popular markets for US hemp growers to pursue, yet practical agronomic information is still limited in geographic and temporal scope, especially for organic production systems. Despite limited understanding of best practices for cultivation, farmers have shown a high level of interest, with one study finding 85% of respondents wanting more information on how to incorporate hemp into their farms [3]. For growers that have incorporated hemp into their operations, many practical and regulatory questions remain unanswered especially from unbiased sources [4]. Fertility practices are one area where there is differing and possibly detrimental information circulating among growers, demonstrating the need for more research and clarity.

Adequate supply of macro- and micro-nutrients is essential for efficient and sustainable cultivation of any crop. The most important nutrients for plants are nitrogen (N), phosphorus (P), and potassium (K). Little is currently known about specific fertility input needs for hemp, but it does require supplemental fertilizer to maximize yield [5]. Nitrogen

fertilization is especially important during the first month of growth, as young hemp plants establish [6]. As plants continue to develop, adequate nitrogen is needed to ensure photosynthetic pigment biosynthesis, carbon fixation, and water relation homeostasis [7]. Phosphorus is also needed throughout the entire life cycle of hemp. Phosphorus plays a key role in disease resistance and plant health, and hemp plants deficient in phosphorus will become stunted, resulting in lower yields [6]. There are few studies on hemp potassium requirements and, so far, they have been inconclusive [8]. In drug type cannabis production, some growers believe that cannabinoid yields can be optimized by manipulating specific aspects of fertility, yet this has not been rigorously tested [8]. However, this belief has left hemp farmers unsure if different fertility practices will alter cannabinoid content, potentially leading to noncompliant crops.

In the United States a hemp crop becomes noncompliant when it exceeds a threshold of 0.3% THC on a dry weight basis. Noncompliant fields must be destroyed or remediated, leaving the grower with a net loss of income. Due to the risk of crop loss, growers are hesitant when applying fertilizer, regardless of increase in yield, thanks to unsubstantiated claims about increased fertility correlating with altered cannabinoid production. In Wisconsin, during the first legalized production season in 2018, many growers shared fears that more intensive fertilization would increase the risk of noncompliance, particularly if using fertility regimes developed for drug-type cannabis production [9]. Initial research has found the relationship between cannabinoids, fertility, and biomass appears to be complicated and inconsistent. In 2019, Bernstein et al. found the concentrations of the major cannabinoids THC and CBD in the topmost plant flowers were not affected by increased Phosphorus and NPK treatments. However, a 16% reduction of THC concentration was observed in the inflorescence leaves and NPK supplementation increased CBG levels in flowers by 71%, and lowered CBN levels in both flowers and inflorescence leaves [10]. A different evaluation on the effect of organic fertilization in cannabis during the vegetative growth stage suggested that over-fertilization may decrease both THC concentration in floral material and floral dry weight [11]. Similarly, Anderson II et al. found negative correlations between cannabinoid concentration and increased fertility due to a dilution effect caused by increases in plant biomass [12].

Understanding how fertility management affects cannabinoid accumulation is imperative to help guide farmers, so they can focus on yield and quality without concern of noncompliance. While initial studies have helped elucidate cannabinoid response to certain nutrient levels, fertility strategies marketed to hemp growers differ markedly beyond modulating nutrient levels, including cost and difficulty of application. For instance, fertilizer products marketed specifically for cannabis often cost significantly more than standard organic fertilizer blends, but this cost may be unwarranted if there is no demonstrated yield boost. Controlled application of fertility is also more challenging in outdoor production as compared to indoor production, but it is where most hemp producers grow and therefore warrants research [13]. A reasonable effort of application and cost to maximize yield is especially important as fertilizer prices rise and hemp biomass prices decrease as the tumultuous hemp marketplace reaches equilibrium.

The intention of this study was not to test different nutrient application rates, but rather to assess the implications of fundamentally different fertility strategies for biomass production and compliance. To address grower concerns that intensive fertilizer practices will cause hemp to become noncompliant, four organic fertility strategies were chosen for this study that represent common approaches found in hemp production. The objective was to determine if (1) different organic fertility treatments change the CBD to THC ratio and thus increase risk of noncompliance and (2) specific organic fertility regimes increase biomass yield more than others. Based on prior research, we hypothesize that fertility treatments would not significantly change cannabinoid ratio, and that higher N fertility regimes would increase biomass yield. By choosing commonly used organic fertility treatments, growers will be able to use the data from this study to inform their fertilizer

choice. Our study results will supply agricultural professionals with information that is practical and applicable to their grower operations.

## 2. Materials and Methods

### 2.1. Site and Treatment Description

Field trials were conducted in Cottage Grove, Wisconsin, USA from May 2019 to October 2019 and May 2020 to October 2020 (43.012303, −89.203072). Two adjacent areas of certified organic land were used for the experiment, both of which had previously been permanent fallow weedy sod since 2012, managed only by mowing with no additional inputs. Soil type was Dodge Silt Loam, with organic matter of 2.3% in 2019 and 2.6% in 2020, and a soil pH of 6.9 in 2019 and 6.6 in 2020. Phosphorous levels tested extremely high in soil test results (56 ppm in 2019, 51 ppm in 2020) from 15 cm soil cores. The experiment was established as a randomized complete block design with three replications, with fertility treatment as the main factor and sampling date as a sub factor. Fertility treatments (Table 1) were chosen in discussion with advising growers to be representative of ingredients and practices commonly available to and used by organic hemp growers in Wisconsin. At the time of study implementation, no peer reviewed nutrient recommendations were available for cannabinoid hemp production. N levels were based on a combination of grower practices, seed company recommendations, and product recommendations (in the case of the BQ treatment), while P and K levels were based on high demand crop requirements according to University of Wisconsin Extension guidelines (Laboski and Peters, 2019), due to high background levels of P and K seen in soil tests and concerns with overapplication. Fertilizer treatments included: (1) no fertility added (NA); (2) manure-based compost (CM); (3) a standard organic fertilizer blend including composted poultry manure, feathermeal, soft rock phosphate, and sulfate of potash (ST); and (4) an intensive or 'boutique' fertility treatment designed for recreational or medical production of drug-type cannabis, with ingredients including composted poultry manure, feathermeal, sulfate of potash, alfalfa meal, kelp meal, neem seed meal, azomite, Tiger 90 Sulfate, compost, and two liquid fertility products for ongoing fertigation (BQ). There were five plants per plot, and the two edge plants were treated as guard plants to provide an additional buffer between fertility treatments. All data were collected from the middle three plants of each plot. Additional guard rows were planted at the perimeter. The experimental area was managed with 4' black plastic mulch covering the planting bed, and a living mulch of annual ryegrass (*Lolium multiflorum*) and crimson clover (*Trifolium incarnatum*) between beds.

### 2.2. Field Activities

Sod was terminated with a 1.83 m wide PTO-driven rotovator three weeks prior to planting. One week prior to planting, fertilizer was zone applied in the center 0.61 m of each planting bed and incorporated using a walk-behind Grillo model 870 tractor and 0.76 m rototiller. Plastic mulch and drip irrigation were applied, and living mulch was seeded in the aisles at a rate of 15.57 kg ha$^{-1}$ crimson clover and 31.37 kg ha$^{-1}$ annual ryegrass. The hemp cultivar Abacus™ (Colorado CBD Seeds; Loveland, CO, USA) was selected and seedlings grown from feminized seed were hand transplanted at three weeks old on 18 June 2019, and four weeks old on 23 June 2020, with a between row spacing of 1.83 m and in-row spacing of 1.52 m. Irrigation was applied as needed. Planting holes were hand-weeded and living mulch was mowed at a 15 cm height on an approximate weekly basis. The BQ treatment received two applications of Buddha Grow in July and two applications of Buddha Bloom in August.

**Table 1.** Organic fertility treatments and products.

| Treatment | Year | Item and Source | Application Rate (h$^{-1}$) | N | P | K |
|---|---|---|---|---|---|---|
| | | | | kg ha$^{-1}$ | | |
| None (NA) | 2019, 2020 | - | 0 | - | | |
| Compost (CM) | 2019 | Compost [1] | 113 m$^3$ | 107 | 333 | 314 |
| | 2020 | Compost [2] | 113 m$^3$ | 204 | 148 | 122 |
| Standard (ST) | 2019, 2020 | TOTAL | | 158 | 17 | 91 |
| | | Composted Poultry Manure (CPM) [3] | 415 m$^3$ | 8 | 17 | 12 |
| | | Feathermeal [3] | 1148 kg | 149 | 0 | 0 |
| | | Sulfate of Potash [3] | 156 kg | 0 | 0 | 78 |
| Boutique (BQ) | 2019 | TOTAL (2019) | | 228 | 230 | 224 |
| | 2020 | TOTAL (2020) | | 266 | 77 | 147 |
| | 2019 | Compost [1] | 45 m$^3$ | 43 | 211 | 126 |
| | 2020 | Compost [2] | 45 m$^3$ | 82 | 59 | 49 |
| | 2019, 2020 | CPM [3] | 220 kg | 4 | 9 | 7 |
| | | Feathermeal [3] | 976 kg | 127 | 0 | 0 |
| | | Sulfate of Potash [3] | 122 kg | 0 | 0 | 61 |
| | | Alfalfa Meal [4] | 122 kg | 3 | 0.6 | 3 |
| | | Kelp Meal [4] | 488 kg | 5 | 0.6 | 10 |
| | | Neem Seed Meal [4] | 732 kg | 44 | 7 | 15 |
| | | Granulated Azomite [5] | 488 kg | 0 | 0 | 1 |
| | | 90% Sulfate [6] | 28 kg | 0 | 0 | 0 |
| | | Buddha Grow and Bloom [7] | 35 kg | 2 | 2 | 2 |

[1] Cowsmo Inc.—Cochrane, WI, USA; [2] University of Wisconsin—Madison, West Madison Agricultural Research Station—Verona, WI, USA; [3] Cashton Farm Supply—Cashton, WI, USA; [4] Down to Earth Distributors—Eugene, OR, USA; [5] Azomite Mineral Products—Nephi, UT, USA; [6] Tiger-Sul Products—Atmore, AL, USA; [7] Roots Organics—Aurora Innovations—Eugene, OR, USA.

Beginning the first week of flowering, 5 cm of apical flower tissue was selected from three flowering plants per treatment and dried in brown paper bags at 43 °C for 48 h. Flower sampling was conducted in accordance with the USDA Hemp Final Rule. Sampling was conducted on a weekly basis until overnight temperatures were projected to drop below a killing freeze at −2 °C. Tissue was processed for cannabinoid content analysis using HPLC by the University of Wisconsin, Madison, Wisconsin Crop Innovation Center (Madison, WI, USA) in 2019. For each sample, 20 mg of dried, milled tissue was mixed with 1.0 mL extraction solvent (8:2 acetonitrile to methanol) in a 1.5 mL centrifuge tube, containing a ceramic homogenizer, by high-speed shaking at room temperature with a vortex for 10 min. Each sample was then centrifuged for 10 min at 3000 RCF and 125 μL was transferred to an amber microcentrifuge vial containing 875 μL of extraction solvent (a 1/8 dilution). Samples were filtered using a 0.45 μm regenerated cellulose filter vial and subjected to HPLC analysis (Agilent 1290 Infinity II) using an InfinityLab Poroshell 120 EC-C18 3 x 100 mm column heated at 35 °C. Samples were injected and eluted at a flow rate of 1.0 mL/minute with a changing gradient to match elution: 30:70 water:acetonitrile (0.05% formic acid in both) to 28:72 water:acetonitrile at 1.95 min then an immediate switch to 22:78 water:acetonitrile to 18:82 water:acetonitrile at 4.00 min, ending with 100% acetonitrile at 5.00 min. Absorbance was measured at 220 nm. The following standards were used as calibrants: Tetrahydrocannabinolic acid (THCA), Δ9-THC, cannabidiolic acid (CBDA), CBD, cannabichromenic acid (CBCA), cannabichromene (CBC), cannabigerolic acid (CBGA), cannabigerol (CBG), cannabinol (CBN), and Δ8-THC (Sigma Aldrich, St. Louis, MO, USA). Cannabinoid content was calculated as Total CBD = CBD + (CBDA × 0.877) and Total THC = Δ9-THC + (THCA × 0.877) in accordance with the USDA Hemp Final Rule. In 2020, cannabinoids were quantified by a private testing facility, Rock River Laboratories (Watertown, WI, USA), using the same compliance testing procedure licensed by the Wisconsin Department of

Agriculture, Trade, and Consumer Protections (facility change necessitated by COVID-19 pandemic restrictions). Whole plants were harvested on 16 October 2019 and 20 October 2020. Plants were dried with forced air for seven days at 43 °C, after which floral biomass was removed from main stems by hand and weight was taken when constant moisture was observed.

### 2.3. Data Analysis

Data were analyzed in R (R.app GUI 1.73, 7892 Catalina build, S. Urbanek & H.-J. Bibiko, ©R Foundation for Statistical Computing, 2020). ANOVAs for cannabinoid content were run using the aov() function in the stats package (R Core Team 2020) using the following model:

$$Y_{ijkl} = A_i + \underline{B_{j(i)}} + WP_k + AWP_{ij} + TP_d + ATP_{id} + WPTP_{kd} + AWPTP_{ikd} + \epsilon_{ijkl} \quad (1)$$

where $A_i$ = year i (i = 2019, 2020),
$\underline{B_{j(i)}}$ = Block j (j = 1, 2, 3) within year i,
$\overline{WP_k}$ = whole plot factor for fertility treatment k (k = NA, CM, ST, BQ),
$TP_d$ = timepoint factor (d = 1, 2, 3, 4, 5, 6, 7), and
$\epsilon_{ijkl}$ = the residual error of the i-th year, j-th block, k-th fertility treatment, and d-th timepoint.

Year, fertility treatment, and timepoint are considered fixed effects, and block was considered a random effect. Normality and equality of variances were checked visually using normal-quantile plots of real data and residuals, residuals and fitted values plots, and with shapiro_test() and levene_test() functions in the rstatix package [14]. All data adequately met assumptions except for ST and BQ groups in 2020 for cannabinoid ratio data. However, transformations did not improve assumptions, and tests were performed on original data given they were close to meeting assumptions, balanced relative to other groups within that year, and ANOVAs are relatively robust to violations of assumptions of normality and equality of variance. When ANOVAs were significant, the emmeans() function in the emmeans package was used for estimating Least-Squares Means and multiple comparisons of means, using the Tukey method for adjusting *p*-values when comparing a family of multiple estimates [15]. ANOVAs for biomass yield data followed the same procedure but the model did not include timepoint factors or associated interactions.

### 3. Results

### 3.1. Weather

Weather deviated slightly across years and as compared to the 40-year average (Table 2). Notable weather exceptions included above average rainfall in 2019 and less rainfall than average during the 2020 production season. Both years also had more Growing degree day unit (GDDU) accumulation than a typical Wisconsin season.

**Table 2.** Weather data from the Madison Dane County Regional Airport weather station (~18 km from the study site) showing seasonal data during the course of the study, and deviations from 40-year averages (1979–2020).

| Period | Total Precipitation in cm (Deviation from 40 Yr Average) | Average Daily Temperature in C (Deviation from 40 Yr Average) | * GDD50 Accumulation (Deviation from 40 Yr Average) |
|---|---|---|---|
| 1 June 2019–31 October 2019 | 83 (+28) | 17.36 (+0.14) | 2628 (+129) |
| 1 June 2020–31 October 2020 | 70 (−15) | 17.06 (−0.20) | 2599 (+100) |

* Growing degree day units (GDDU) are calculated by subtracting a base temperature, in this case 50 °F, from the average daily air temperature as a way to quantify the potential for insect or plant development or growth [16].

### 3.2. Biomass Yield

Per plant biomass ranged from 263 g to 1186 g in 2019 and 320 g to 1145 g in 2020 (Figure 1, Table S1). Biomass yield was clearly affected by fertilizer treatment, and there was also a significant fertilizer × year interaction (Table 3, Figure 1). Across both years, ST and BQ yielded higher than NA ($p < 0.05$), and there was weak evidence that CM yielded higher than NA ($p < 0.1$). In 2019, BQ yielded more than CM and NA ($p < 0.05$) while other pairwise treatment combinations were not significantly different from each other. In 2020, CM yielded higher than NA ($p < 0.05$), and other treatment combinations were not differentiated. Only CM had clearly different yield between years, with higher yield in 2020 when the more N rich compost source was used ($p < 0.01$).

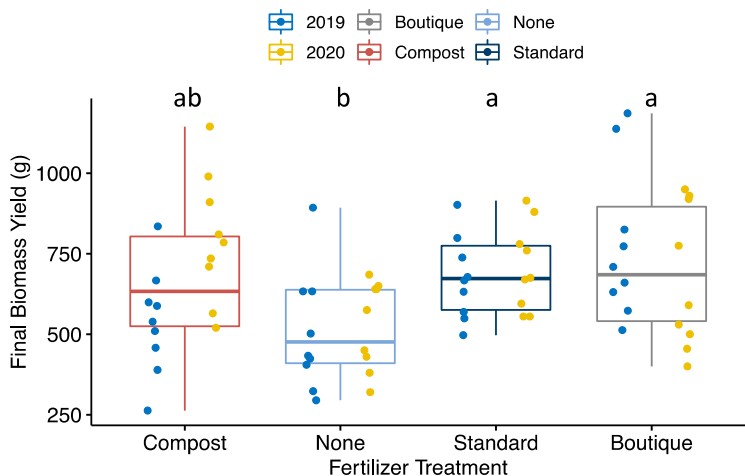

**Figure 1.** Biomass yield per plant (g) separated by fertility treatment, represented by blue dots for 2019 and yellow dots for 2020. Boxplots show the interquartile range of biomass across both years.

**Table 3.** Biomass yield per plant (g). Fertilizer treatments with the same uppercase letter under yield were not significantly different ($p < 0.05$) across treatments for results averaged over the levels of block and year. Lowercase letters indicate significance groupings for the simple main effect of fertilizer treatments within one year (lowercase letters in yield column), or the simple main effect of year within one fertilizer treatment (year column). Results for Type III ANOVA with Satterthwaite's method are included.

| Treatment | Year | Biomass Yield per Plant (g) |
|---|---|---|
| None (NA) | Overall | 517.28 [B] |
| | 2019 | 504.56 [b] |
| | 2020 | 530.00 [b] |
| Compost (CM) | Overall | 667.67 [AB] |
| | 2019 [b] | 538.67 [b] |
| | 2020 [a] | 796.67 [a] |
| Standard (ST) | Overall | 689.78 [A] |
| | 2019 | 670.11 [ab] |
| | 2020 | 709.44 [ab] |
| Boutique (BQ) | Overall | 725.44 [A] |
| | 2019 | 778.67 [a] |
| | 2020 | 672.22 [ab] |
| Significance: | | |
| Year: | | ns |
| Fertilizer: | | F = 6.68, $p < 0.01$ |
| Year × Fertilizer | | F = 3.18, $p < 0.05$ |

### 3.3. Cannabinoid Accumulation Rates and Ratio

Total CBD and Total THC concentration ranged from 7.23% to 15.26% and 0.21% to 0.57% in 2019, and 4.28% to 11.7% and 0.16% to 0.40% in 2020 (Table S2). CBD and THC accumulation were both lower in 2020 than in 2019 (Figure 2). Accumulation of both cannabinoids was also affected by fertilizer and sampling timepoint. There were two-way interactions between fertilizer and year, sampling timepoint and year, and fertilizer and timepoint (Table 4). The interaction between sampling timepoint and year was driven by lower overall accumulation in 2020. Within each level of sampling date, fertilizer treatments were only clearly different across both years in the 4th and 6th timepoints, with BQ having lower CBD concentrations than CM in the 4th timepoint, and both CM and NA in the 6th timepoint. The interaction between fertilizer and timepoint for THC content was driven by clearly lower concentrations in BQ than other treatments within timepoint 6, while fertilizers were not significantly different within other timepoints. Overall, BQ had lower cannabinoid concentrations than CM ($p < 0.01$), but that was caused by strong differences from CM in 2019 (when BQ also accumulated significantly less cannabinoid content than CM and ST); in 2020 there were no clear differences between treatments.

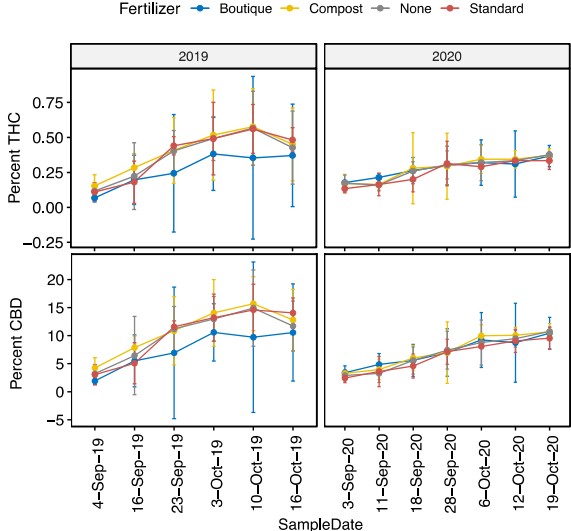

**Figure 2.** Total THC and total CBD over sampling time, separated by fertilizer treatment in 2019 and 2020.

**Table 4.** Cannabinoid content. Fertilizer treatments with the same uppercase letter in a given column were not significantly different ($p < 0.05$) across years for results averaged over the levels of block within year. Results for Type III ANOVA with Satterthwaite's method are included.

| Treatment | %CBD | %THC | CBD:THC Ratio |
|---|---|---|---|
| None (NA) | 8.03 [AB] | 0.31 [AB] | 24.9 |
| Compost (CM) | 8.82 [A] | 0.33 [A] | 25.9 |
| Standard (ST) | 8.06 [AB] | 0.31 [AB] | 25.3 |
| Boutique (BQ) | 6.99 [B] | 0.27 [B] | 26.1 |
| Significance: | | | |
| Year: | F = 9.90, $p < 0.01$ | F = 6.96, $p < 0.05$ | F = 7.02, $p < 0.05$ |
| Fertilizer: | F = 7.35, $p < 0.001$ | F = 6.53, $p < 0.01$ | ns |
| Year × fertilizer | F = 3.36, $p < 0.05$ | F = 2.76, $p < 0.1$ | ns |
| Timepoint | F = 2.76, $p < 0.05$ | F = 2.78, $p < 0.05$ | F = 2.58, $p < 0.05$ |
| Year × timepoint | F = 5.87, $p < 0.01$ | F = 8.74, $p < 0.0001$ | F = 11.42, $p < 0.0001$ |
| Fertilizer × timepoint | F = 2.16, $p < 0.01$ | F = 2.92, $p < 0.001$ | F = 3.72, $p < 0.0001$ |
| Year × timepoint × fertilizer | ns | ns | ns |

The only difference in THC or CBD accumulation was driven by the lower concentrations in the BQ treatment than all other treatments in 2019 ($p < 0.01$), while there were no differences between treatments in 2020. However, the lower concentrations for BQ in 2019 were seen for both CBD and THC. Accordingly, there was no clear effect on CBD:THC ratio from fertilizer treatment ($p > 0.1$, F = 2.0), or interactions between fertilizer with year ($p > 0.2$, F = 1.49) or sampling date ($p > 0.9$, F = 0.47). There was a significant timepoint × year interaction for cannabinoid ratio ($p < 0.0001$, F = 13.53), because the ratio did not differ over time in 2019, while in 2020 the measured ratio started significantly lower than it ended. In 2020, there were differences between the first timepoint, the 2nd–4th timepoints, and the last three timepoints (Figure 3).

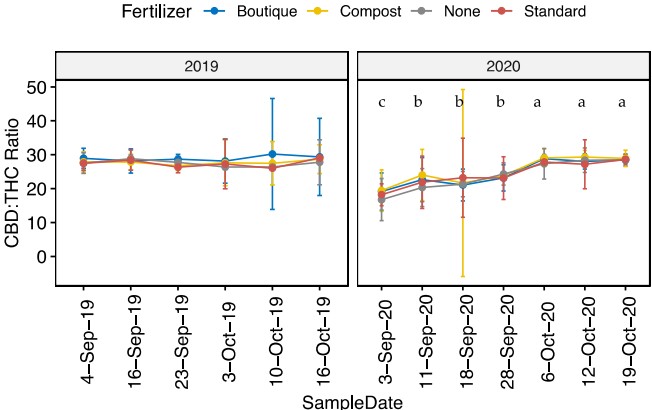

**Figure 3.** CBD:THC ratio over sampling time, separated by fertilizer treatment in 2019 and 2020. Timepoints with the same lowercase letter were not significantly different ($p < 0.05$) for results averaged over the levels of fertilizer and block.

## 4. Discussion

As hypothesized, we saw a consistent biomass yield benefit from more intensive, higher N fertilization regimes. Both the ST and BQ fertilizer blends performed comparably, yielding significantly more floral biomass than the control treatment (NA). The effect of CM was more variable, yielding similarly to the control and clearly worse than BQ in 2019, but significantly more than the control in 2020, ranking highest that year.

Our variable results reinforce two possible risks: first, the inherent variability of compost as a sole source of fertility due to supply fluctuations and different sources which may have markedly different nutrient contents; and second, the overarching risk that environmental factors play a strong role in determining yield, regardless of fertilizer. For instance, the yield benefit of ST was consistent, leading it to rank highly across both years, despite not being significantly different from other treatments in an individual year. On the other hand, the overall yield increase seen in BQ as compared to the control was driven by significant differences in 2019, while in 2020 BQ ranked lower for yield than both ST and CM. Growers may want to prioritize consistency over large yield swings year over year.

The key question that inspired this study was whether the CBD to THC ratio would change based on fertilizers applied. Our field results nicely compliment the Bernstein et al. (2019) and Anderson II et al. (2021) greenhouse studies, and we found altered fertility may change cannabinoid abundance but does not appear to alter the CBD to THC ratio in floral biomass [10,12]. While controlled environment studies are critical for informing production in certain markets, particularly medical and recreational cannabis, other markets focused on cannabinoid production, such as CBD, may not have the margin to afford controlled environment production. Open field environments are inherently more prone to variability than greenhouse studies where the bulk of cannabinoid research has taken place. While both years accumulated more GDDUs than average, 2019 also had higher rainfall, which may have changed the nutrient dynamics. Since hemp has been documented to have the highest demand for N during the first month of development, the additional

moisture likely contributed to more rapid N cycling that may have spurred critical early growth [6,17]. Although previous research has suggested that environmental stressors do not change cannabinoid ratio, future studies are needed for a range of hemp market types, cropping systems, and intensities [18]. CBD:THC ratios may also appear to fluctuate across development due to challenges in early timepoint cannabinoid determination—when cannabinoid abundance is low concentrations may be less accurate due to HPLC limits of detection and quantification. More studies such as Darby et al. (2019) on fertility approaches in outdoor production environments are needed to continue to investigate the interaction between environmental conditions and plant nutrient dynamics over time [19].

Interestingly, although CBD:THC ratios showed remarkable stability between treatments within each date, there were significant differences in cannabinoid concentration in our study driven by the lower total cannabinoid concentrations in the BQ fertility treatment in 2019, the same year it had the highest yield. This may point to tradeoffs in maximizing vegetative growth and subsequent biomass yield through more intensive fertilization as contrasted with maximizing cannabinoid concentration. Results from Tang et al. (2016) suggest there may be tradeoffs between hemp seed and stem yield based on relative time to maturity [20]. Future studies for cannabinoid market classes should strive to include measurements of relative vegetative vigor and senescence in relation to fertilizer rate to assess whether fertilizer impacts maturation time and whether there are subsequent tradeoffs for cannabinoid concentration and floral biomass yield. Understanding such tradeoffs will be important for growers seeking to refine their production management.

Overall, studies informing best practices for fertilizer application in hemp production for cannabinoids have continued to be sparse. Growers have had to move forward using existing, familiar fertilization strategies. While greenhouse studies are crucial for accurately characterizing the potential influence of nutrients on cannabinoid accumulation, studies in outdoor production environments are crucial for informing grower recommendations. For instance, the relatively high background P and K levels at our production site may have overshadowed any effects seen of the supplementary nutrients in the BQ treatment, as opposed to the significant effect seen by Bernstein et al. (2019) in a greenhouse with conventional fertilizers [10]. Furthermore, our study aimed to investigate fertilizer approaches representative of typical grower practices.

Fortunately for growers, time- and money-intensive boutique blends marketed specifically for cannabis may not be necessary to have a high yielding crop. With that said, fertilizer should be used, as there was a clear difference in yield between the control and both BQ and ST, although more inherently variable sources of fertility, such as compost, should be considered with caution. Given the rising cost of inputs, it is also important for growers to know that basic fertilizer blends can be just as effective as more expensive fertility programs marketed specifically to cannabis growers. Our results show that additional organic fertilizer with higher nitrogen levels will not cause lower CBD to THC ratios. Based on the fundamentally different fertility approaches assessed here, our results suggest that growers should be able to confidently use common organic fertilizers to increase floral biomass yield without increasing their risk of noncompliance.

## 5. Conclusions

Coupled with the excitement surrounding hemp, there is also uncertainty and risk, with major gaps in research, extension, and education due to previous legal barriers. These gaps are even greater in organic production systems; in a 2020 survey of hemp producers, nearly 75% of respondents identified research in organic production systems as very or extremely important [4]. Most of the previous research on hemp fertility needs has focused on N fertilization in fiber and grain production, or on greenhouse production utilizing conventional synthetic fertilizers [8]. Organic producers need research-based recommendations for hemp fertility management across a wide range of production systems and end products. Producers should also feel comfortable selecting fertility options based on yield without worrying about potential for producing a noncompliant crop. Lastly,

cannabis is frequently produced in high-input and environmentally unsustainable systems, and there is a need to develop sustainable, profitable, and equitable best practices moving forward. Continued research into optimized fertility management will be imperative for the successful reintroduction of hemp into 21st century agriculture.

**Supplementary Materials:** The following supporting information can be downloaded at: https://www.mdpi.com/article/10.3390/su14106222/s1, Table S1: Final weight of hemp biomass (grams) for four fertility treatments in years 2019 and 2020. Table S2: Total CBD, total THC, and CBD:THC ratios for four fertility treatments across seven timepoints in years 2019 and 2020.

**Author Contributions:** Conceptualization, D.B.; methodology, D.B. and S.E.; formal analysis, D.B., G.C. and S.E.; writing—original draft preparation, D.B., G.C. and S.E.; writing—review and editing, D.B., G.C. and S.E.; funding acquisition, D.B. and S.E. All authors have read and agreed to the published version of the manuscript.

**Funding:** This research was funded by the UW-Madison Division of Extension and UW-Madison CALS Industrial Hemp Capacity Grant.

**Institutional Review Board Statement:** Not applicable.

**Informed Consent Statement:** Not applicable.

**Data Availability Statement:** Data are contained within the article or supplementary material.

**Acknowledgments:** We thank Aaron Barton, Jacy Swiggum, Leah Sandler, Senay Ugur, Sean Kim, and Anna Skye Bruce for technical assistance, Randy Kohn from West Star Organics for providing certified organic land and compost, and Keegan Murray-King and Circadian Organics for providing fertilizers, transplants, and expertise. We also thank the team at the Wisconsin Crop Innovation Center and Rock River Laboratory for assisting with cannabinoid quantification. We further thank Tim Erdman for a donation that made this work possible.

**Conflicts of Interest:** The authors declare no conflict of interest.

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
