# Peer review of "Different Fertility Approaches in Organic Hemp (Cannabis sativa L.) Production Alter Floral Biomass Yield but Not CBD:THC Ratio"

_sustainability, doi:10.3390/su14106222_

Round 1

Reviewer 1 Report

The manuscript is well written with good structure, however authors are advised to add some illusterative figures

ore over, they should drew a conclusion on their work , as i did not see a conclusion section 

Author Response

The manuscript is well written with good structure, however authors are advised to add some illusterative figures

more over, they should drew a conclusion on their work , as i did not see a conclusion section 

Thank you for your comments. We have added additional information in Table 3 and created a new table (Table 4) to summarize our results. We have also added a conclusion section (L560-575).

Author Response

Thank you for your comments. Please see the attached document for our response.

Reviewer 3 Report

In my opinion, this paper may be interesting for reader of  Sustainability Journal. Nevertheless, there are some items that should be addressed before acceptance:

THE TITLE OF MANUSCRIPT SHOULD BE IMPROVED

However, the manuscript is framed nicely.

In keyword section Cannabis sativa should be in Italics, check in whole manuscript botanical name should be in italics

In the abstract section we and our results / findings not acceptable. Improve the sentence

Improve table 1 (Line No. 201)

Reframe the tables number (there is 2 table with 1 number)

Font of references is not as per the journal.

Introduction:

The introduction clearly state the purpose of the research.

Results

There are no major reservations to the results. The results are appropriately described. 

Discussion

The Discussion is well structured; the research results are confronted with other authors’ research.

Conclusions

 The conclusions are too general. Please make them more specific.

Author Response

In my opinion, this paper may be interesting for reader of Sustainability Journal. Nevertheless, there are some items that should be addressed before acceptance:

THE TITLE OF MANUSCRIPT SHOULD BE IMPROVED SE

We have updated the manuscript title.

However, the manuscript is framed nicely.

In keyword section Cannabis sativa should be in Italics, check in whole manuscript botanical name should be in italics

This has been updated throughout the manuscript.

In the abstract section we and our results / findings not acceptable. Improve the sentence

We have removed “we” and “our” from the abstract.

Improve table 1 (Line No. 201)

Table 1 has been improved.

Reframe the tables number (there is 2 table with 1 number)

Thank you for catching our mistake. The tables and figures have been double checked and mismatched ones have been changed.

Font of references is not as per the journal.

Font for references has been changed from 10pt to the correct font size of 9pt.

Introduction:

The introduction clearly state the purpose of the research.

Results

There are no major reservations to the results. The results are appropriately described. 

Discussion

The Discussion is well structured; the research results are confronted with other authors’ research.

Thank you.

Conclusions

The conclusions are too general. Please make them more specific.

We have added a concluding paragraph specific to our findings (L560-575)